

# Electrodynamic balance–mass spectrometry of single particles as a new platform for atmospheric chemistry research

Adam W. Birdsall[1], Ulrich K. Krieger[2], Frank N. Keutsch[1,3]

[1]Department of Chemistry and Chemical Biology, Harvard University, Cambridge, MA, USA
[2]Institute for Atmospheric and Climate Science, ETH Zürich, 8092 Zürich, Switzerland
[3]School of Engineering and Applied Sciences, Harvard University, Cambridge, MA, USA

*Correspondence to*: Frank Keutsch (keutsch@seas.harvard.edu) and Ulrich Krieger (ulrich.krieger@env.ethz.ch)

New analytical techniques are needed to improve our understanding of the intertwined physical and chemical processes that affect the composition of aerosol particles in the Earth's atmosphere, such as gas–particle partitioning and homogenous or
heterogeneous chemistry, and their ultimate relation to air quality and climate. We describe a new laboratory setup that couples an electrodynamic balance (EDB) to a mass spectrometer (MS). The EDB stores a single laboratory-generated particle in an electric field under atmospheric conditions for an arbitrarily long length of time. The particle is then transferred via gas flow to an ionization region that vaporizes and ionizes the analyte molecules before MS measurement. We demonstrate the feasibility of the technique by tracking evaporation of polyethylene glycol molecules and finding agreement
with a kinetic model. Fitting data to the kinetic model also allows determination of vapor pressures to within a factor of 2. This EDB-MS system can be used to study fundamental chemical and physical processes involving particles that are difficult to isolate and study with other techniques. The results of such measurements can be used to improve our understanding of atmospheric particles.

## 1 Introduction

Aerosol particles in the Earth's atmosphere affect both the planet's climate system and human health (Boucher et al., 2013; Lelieveld et al., 2015). Because of these twin impacts, one long-standing goal of atmospheric research has been to assemble via experiment a detailed fundamental understanding of the coupled chemical–physical processes controlling the prevalence and composition of these particles, such as gas-particle partitioning (reviewed in Bilde et al., 2015), homogeneous and heterogeneous chemistry (e.g., George et al., 2015; Herrmann et al., 2015; Kroll et al., 2015), and kinetic barriers arising
from high particle viscosity or phase separation (e.g., Bastelberger et al., 2017; Shiraiwa et al., 2013).

One avenue of research directed toward that goal has been to study the behavior of individual model aerosol particles under controlled laboratory conditions. Researchers have studied particles deposited onto a substrate, or alternately, particles levitated by means of a "trapping" force originating from an electric field, radiation pressure of a laser beam, or acoustic waves. Levitated droplet experiments are appealing because they mimic aerosol particles in the ambient environment in
certain key ways: the presence of a surrounding bath gas, an enhanced surface-to-bulk ratio, the absence of physical contact



with a substrate, and the ability to study supersaturated particles. Multiple laboratories have analyzed levitated droplets using optical techniques, which can be used to track evaporation and condensation via highly precise particle sizing, as well as some changes in chemical composition (reviewed in Krieger et al., 2012).

Due to the high chemical complexity of aerosol particles in the atmosphere, an analytical system for levitated particle
experiments providing greater chemical specificity than existing optical methods is desired. Mass spectrometry can help fill that need. One laboratory has used a newly-developed branched quadrupole trap (BQT) design, which suspends particles with diameters on the order of microns or 10s of microns within an electric field, to obtain mass spectra of analyte droplets ejected from the BQT using a paper spray ionization source (Jacobs et al., 2017). Among other features, the BQT design lends itself to study of condensed-phase reactions, triggered by the coalescence of two droplets of differing composition with
sub-millisecond mixing times. Previous work has also reported measuring mass spectra of aqueous droplets suspended in acoustic traps (Crawford et al., 2016; Stindt et al., 2013; Warschat et al., 2015; Westphall et al., 2008). The aqueous droplets suspended in acoustic traps tend to have a diameter on the order of a millimeter, much larger than the micron or submicron diameter of atmospheric aerosol particles. "Online" monitoring of the droplet's composition while the droplet is in the trap has been achieved with these systems, though in some cases the droplet needs to make contact with a physical support while
ionization is occurring, due to the disruptive impact of the ionization source on the trapping potential. Another line of research has measured the "offline" mass spectra of levitated micron-sized particles after deposition onto a substrate, using laser desorption ionization techniques (Bogan and Agnes, 2002; Haddrell and Agnes, 2004; Haddrell et al., 2005).

Other online mass spectral measurements of single aerosol particles, albeit not of levitated particles, have been performed with single particle mass spectrometers (SPMS). The instruments size micron or submicron aerosol particles based on the
terminal velocity after acceleration, and then collect mass spectra on a single-particle basis, with ionization typically achieved via laser desorption (reviewed in Pratt and Prather, 2011). Generally the amount of fragmentation induced by the laser desorption ionization makes identification of single organic analyte ions difficult, but instruments such as the Single Particle Laser Ablation Time-of-Flight mass spectrometer (SPLAT) have used a two-step laser desorption technique to generate mass spectra with a small enough degree of fragmentation, and enough reproducibility, such that organic analyte
molecules can be identified (Zelenyuk et al., 2009). Such an instrument has been used to study, in laboratory chamber experiments lasting on the timescale of hours, processes such as evaporation kinetics and the interactions between primary and secondary organic aerosol (Vaden et al., 2011, 2010).

Here we describe a newly-developed system that couples an electrodynamic balance (EDB), which levitates aerosol particles with diameters on the order of 10–30 micrometers for an arbitrarily long amount of time, with mass spectral analysis of the
entire particle. In contrast with the acoustic trap–mass spectroscopy experiments, but similarly to the BQT, our system spatially separates the particle levitation chamber from the mass spectral analysis, meaning the measurement of a single particle destroys that particle, and corresponds to a single residence time in the EDB. The chemical trajectory of how a particle of a given composition transforms is traced out by collecting a series of mass spectra for a set of particles with identical starting composition but varying residence time in the EDB before transfer to the mass spectrometer. As a set of



proof-of-concept experiments, we have analyzed particles containing mixtures of short-chain polyethylene glycol (PEG) molecules. In this paper we demonstrate the ability of the coupled electrodynamic balance–mass spectrometer (EDB–MS) system to measure and quantify on a relative basis the constituent molecules of a multicomponent aerosol particle. The evaporation rates of PEG molecules are shown to agree with a kinetic model of particle evaporation, using literature vapor

pressures, as well as constrain vapor pressures to within a factor of 2 by fitting the model to collected data. We then discuss possible improvements to the experimental system as well as future experiments with this system that leverage the ability of the EDB to trap particles indefinitely to study chemical transformations of aerosol particles over their multiday atmospheric lifetime. For instance, with this system it should be possible to study evaporation in complex nonideal mixtures, and aerosol aging that is not sped up by operating at high reactant concentrations.

**2 Experimental**

**2.1 Design of system**

Figure 1 provides a schematic of the system. The electrodynamic balance (EDB) was previously designed and built at ETH Zurich and has been described elsewhere (Colberg, 2001). In brief, the EDB follows a "double-ring" design in which the electric field trapping the particle originates from a pair of rings acting as high-voltage AC electrodes and two center-drilled

endcaps maintaining a DC potential (Davis et al., 1990). The particle originates from a droplet-on-demand generator based on a commercial inkjet printer cartridge (Hewlett–Packard 51633M) and is then charged inductively by passing through a charged coil. While held in the EDB, the particle is illuminated by a small diode-pumped, solid state laser producing 532 nm light (Lasermate GMA-532-5A9P2) and imaged with a compact CCD camera (JAI CV-A50).

The transfer and ionization source assemblies were newly designed and built at Harvard. The transfer assembly was

constructed of aluminum and stainless steel and kept entirely electrically grounded. Within an outer tube that supports the EDB atop the ionization source, the transfer tube (1/4" OD, length ~14 cm) extends at its top to directly below the lower (grounded) DC endcap of the EDB. A funnel attached to the top of the transfer tube helps reduce turbulence by adapting the inner diameter of the endcap to that of the transfer tube. The transfer tube terminates at its bottom within the ionization source, directly above the vaporization platform.

The ionization source assembly was designed to mount in front of the inlet region of a commercial time-of-flight mass spectrometer (JEOL AccuTOF). The curved face of the assembly's cylindrical housing includes an entrance hole for the transfer tube at the top, two side ports for a 1/2" viewing window and camera, two threaded 1/4" ports to control how much the MS inlet draws in lab air compared to gas from the EDB, and a bottom port that can be used for mounting the vaporization platform or a laser. The flat front face of the housing, opposite the MS inlet, contains a Teflon disc with an

adjustable mount for a 0.30 mm diameter needle used to generate the ionizing corona discharge (typical current through MS orifice plate ~200-300 nA).



The vaporization platform is built around a disk-shaped ceramic positive temperature coefficient (PTC) resistor (TDK B59060) whose temperature self-regulates to 220 °C when a 12 V potential is applied. The resistor is sandwiched between two copper foil electrodes, which in turn are surrounded by two circular glass cover slips (12 mm diameter, 0.14 mm thickness). The stack of materials is secured to a polyether ether ketone (PEEK) base with screws. Upon exiting the transfer

tube, the particle strikes the heated top cover slip and vaporizes. The vapors are drawn immediately into the MS inlet, after first undergoing gas-phase ionization via interaction with the corona discharge.

Electronic control of the EDB system and ionization source was managed via a custom dataflow program (Keysight VEE). The AC voltage for the EDB ring electrodes was generated from a function generator signal (Stanford Research Systems), amplified through a high voltage amplifier (Matsusada). Control, data acquisition, and data analysis from the mass

spectrometer were performed using a commercial software suite (JEOL MassCenter). Relative humidity (RH) and temperature were measured using a combined sensor (Sensirion SHT21) installed in the flow directly upstream of the EDB.

## 2.2 Sizing of levitated particles using the "spring point" method

Immediately after particle introduction into the EDB, "spring point" measurements were made to determine initial diameter (Davis, 2001). The spring point method is based on the equations describing the stability regions of the EDB. These

equations can be shown to relate two parameters that describe the field strength and the drag on the particle at the transition between stable and unstable trapping of a particle—the "spring point". The parameters are related to measured DC amplitude, AC amplitude and frequency, particle diameter, and the "geometrical constant" of the EDB via Eqs. (1) and (2):

$$\beta \times b = \frac{2gV_{ac}}{\omega^2 V_{dc}},\tag{1}$$

$$\alpha = \frac{36\mu}{\rho d_p^2 \omega},\tag{2}$$

where $\beta$ is the field strength parameter; $b$ is the geometrical constant for the specific EDB; $\alpha$ is the drag parameter; $g$ is the gravitational constant (taken as 9.80665 m s⁻²); $V_{ac}$ and $V_{dc}$ are the amplitudes of the AC and DC components, respectively, on the ring electrodes and endcaps; $\omega$ is the angular frequency of the AC component; $\rho$ is the particle's density; $d_p$ is the particle's diameter; and $\mu$ is the viscosity of the surrounding gas (taken as $1.846 \times 10^{-5}$ kg m⁻¹ s⁻¹).

To relate $\alpha$ and $\beta \times b$ at the spring point, single solid polymethyl methacrylate (PMMA) spheres of known 18 μm diameter

(Microbeads AS) were injected into the EDB. The spring point of each sphere was measured for a number of different AC amplitude–frequency combinations, with a total of 22 spring point measurements over 4 different PMMA spheres. From each spring point measurement, $\alpha$ and $\beta \times b$ were calculated using Eq. (1) and Eq. (2), and the data were fit empirically to a second-order polynomial function. This polynomial function was used to convert $\beta \times b$ for each PEG particle, calculated via Eq. (1), to $\alpha$, which in turn was used to calculate a particle diameter via Eq. (2).

We found this method of determining particle diameters to provide values consistent with an alternate calculation method, in which $\alpha$ and $\beta$ are related using the stability curves tabulated in Davis et al. (1990), and $b$ for this EDB is taken to be 2.8 ×





10⁻³ (determined by optimizing the evaporation model fit to data for tetraethylene glycol (PEG-4) evaporation in the polyethylene glycol, average molecular weight 200 (PEG-200), evaporation experiment described below).

## 2.3 Sample preparation

Solutions were prepared using commercially available polyethylene glycol, average molecular weight 200 (PEG-200, TCI),
and monodisperse triethylene glycol (PEG-3), tetraethylene glycol (PEG-4), pentaethylene glycol (PEG-5), and hexaethylene glycol (PEG-6) (99% except PEG-3 97%, Sigma Aldrich). Reliable operation of the inkjet cartridge droplet generator, which requires a liquid with suitable viscosity and surface tension, required all PEG solutions to be dissolved in deionized water. Best performance was found when the PEG was diluted to a weight fraction between 0.20 and 0.30, which optimized the trade-off between consistent droplet generation (sufficiently high concentration of water) and production of larger particles
that were easier to transfer to the ionization assembly (sufficiently high concentration of PEG). Once mixed, samples were pipetted into the well of the inkjet cartridge for particle injection. Because of the dry environment of the EDB in these experiments (<5% RH), effectively all of the water was assumed to evaporate out of the particles after injection on the timescale of seconds, leaving behind a PEG particle with an starting mass proportional to the PEG weight fraction of the prepared solution. Running the evaporation model (described below) with a mole fraction of water of 0.05 (corresponding to
~5% RH, Ninni et al., 1999) confirmed that the presence of water under these dry conditions was predicted to have a negligible effect on the evaporation rate, and hence could be safely disregarded.

## 2.4 Operation of system

A droplet (initial injection volume ~140 pL) was injected from an inkjet cartridge into the electrodynamic balance. The droplet was negatively charged by passing through a coil held at +300 $V_{dc}$. The electric field in the electrodynamic balance
was created from a superposition of an AC field ($V_{pp}$ = 5 kV, $f$ = 100 Hz) and a DC field ($V_{dc}$ = +10 to +20 V).
For particles that were sized, the following procedure was completed within the first two minutes after the droplet was injected into the trap to determine the droplet's spring point: The DC amplitude was adjusted until the droplet was vertically centered in the EDB. The AC frequency was decreased and the AC amplitude was increased (up to 6 kV) until the particle was just at the cusp of no longer being stably trapped. Then, at a fixed AC frequency, the AC amplitude was slowly
increased in 0.01 kV increments until the droplet was observed by eye to no longer be stably trapped in the center of the trapping potential (i.e., when it started tracing a vertically stretched path). The AC adjustment procedure was repeated at a second pair of lower AC frequency and amplitude values when possible. Each trio of DC amplitude, AC frequency, and AC amplitude values allowed for the size of the droplet at that moment to be calculated (Sect. 2.2). The average of the two calculated diameters using the two sets of AC measurements was taken to be the starting diameter of the particle.
Some particles were then immediately ejected from the EDB to the vaporization and ionization region (see ejection procedure below), and hence resided in the EDB for 3 to 5 minutes before mass spectral analysis. Other particles resided in the EDB for longer amounts of time before ejection. For these particles, after the sizing procedure was complete, a 80 sccm





purge flow of nitrogen (Airgas, industrial grade) was introduced from the top of the EDB. The DC trapping voltage was increased to approximately 50 V to keep the droplet near the center of the EDB with this flow. The purge flow remained at this level until droplet ejection.

The droplet ejection procedure started with increasing the nitrogen flow and the DC trapping voltage in tandem so that the droplet's vertical position in the EDB remained constant as the flow increased. It was found that with the current experimental geometry, droplet transfer was most reliable with a nitrogen flow of approximately 200 to 250 sccm and a counterbalancing DC voltage of approximately 200 to 350 V, depending on the droplet mass and charge. Once the flow and voltage were increased, the quarter-inch threaded ports on the ionization region were fully or partially closed (by means of adjustable valves) so that the nitrogen flow into the EDB matched that of the flow entering the mass spectrometer inlet via the transfer tube. The correct extent to close the valves was determined by centering the droplet's horizontal alignment. A centrally aligned droplet was taken to mean the flow out of the bottom of the EDB to the ionization region matched the flow from the top of the EDB. (Horizontal displacement of the droplet was taken as a sign of gas flowing through the EDB's side droplet injection port, due to a mismatch between the nitrogen flow into the top of the EDB and out of the bottom.) Once the flow was set appropriately the DC voltage was switched to 0 V, and the droplet was pulled with the nitrogen flow out of the EDB, down the transfer tube, and onto the vaporization platform in the ionization region.

### 2.5 Quantification

The particle mass spectra were quantified for each mass channel of interest, working at unit m/z mass resolution, using the MS software's "chromatogram" viewer. The height of the peak above the surrounding background, in time, was taken to be the signal strength. The software peak-finding algorithm was used to define the peak height and background, with correct peak identification confirmed by eye. Figure 2 presents a sample time trace of selected ion signals arising from ejection and ionization of a PEG-200 particle.

To account for particle-to-particle variability in MS signal, peaks were normalized to the PEG-6 parent ion signal at 283 m/z. PEG-6 was chosen as an internal standard due to its minimal evaporation over the timescale of these experiments and presence in appreciable amounts. To obtain molar ratios (relative to PEG-6) that can be compared to a model, the normalized peak intensities were then corrected for the molar sensitivity of the specific PEG molecule compared to PEG-6, as determined by measurements of binary droplets of known composition of PEG-6 mixed with PEG-3, PEG-4, or PEG-5 (Table A1).

Mass spectra were also collected for particles consisting of pure PEG-3 through PEG-6 to assess the extent of fragmentation and check for mass coincidence problems. Negligible mass coincidence was found for the parent ion peaks used here, and all molecules were found to have a majority of their signal at the parent ion, with the exception of PEG-3, which had three roughly equal peaks: the parent ion and two fragment ions (Table A2).





### 2.6 Evaporation model

A kinetic model was developed to describe the evaporation of a single PEG droplet levitated in the EDB. The model is initialized with a distribution of PEG components and a particle diameter. The particle is assumed to be an ideal mixture in equilibrium at every instant with the gas phase at the surface, as in Eq. (3):

$$c_{s_i} = X_i \frac{P_{vap_i}}{kT},\tag{3}$$

where $c_{s_i}$ the gas-phase surface concentration of species $i$, $X_i$ is the particle-phase mole fraction of species $i$, $P_{vap_i}$ is the pure component vapor pressure of species $i$ at temperature $T$ inside the EDB, and $k$ is the Boltzmann constant.

The evaporation of species $i$ is then assumed to proceed via Maxwellian flux (Seinfeld and Pandis, 2006), as in Eq. (4):

$$\frac{dn_i}{dt} = 4\pi r D_{g_i}(c_{\infty_i} - c_{s_i}),\tag{4}$$

where $r$ is the particle radius, $D_{g_i}$ is the gas-phase diffusion constant of species $i$, and $c_{\infty_i}$ is the gas-phase concentration of species $i$ at infinite distance from the particle surface (here always taken to be zero). This description of evaporation is strictly true for conditions with no gas flow, whereas we operate with a small nitrogen purge flow (80 sccm) to prevent buildup of PEG vapor within the EDB. However, we conclude our combination of EDB geometry and flow rate leads to a negligible increase in the evaporation rate (Zhang and Davis, 1987).

Parameters used to describe PEG molecules in model calculations are taken from Krieger et al. (2017) and described in Table B1. The model was implemented in Python using the SciPy package's implementation of the LSODA ordinary differential equation solver.

For each experimental data set, the model was run twice as bracketing cases to reflect the uncertainty in literature vapor pressures, as well as particle-to-particle variability in initial diameter and EDB temperature. The slow-evaporation-limit model run used the lowest measured temperature (298.0 K), the largest measured starting particle radius of the particles for a given experiment, and the lower bounds of the literature vapor pressure values (reported as 95% confidence intervals in Krieger et al. (2017)), with the exception of the PEG-6 internal standard, whose vapor pressure was taken as the upper bound of the literature confidence interval. Conversely, the fast-evaporation-limit model run used the highest measured temperature (299.5 K), the smallest measured particle radius, and the upper bounds of the literature vapor pressure confidence intervals, except for the lower bound of PEG-6's vapor pressure. The starting particle radius was typically between 9 and 11 micron, with variability on the order of 10%. The most important contributors to the model output ranges were the uncertainties in vapor pressure and variations in starting radii.

The model's performance was checked by comparison to an experiment performed with a PEG-4 + PEG-6 particle of known starting composition trapped in a similar EDB at ETH Zurich, equipped with a spectrometer that continuously sized the particle via scattering measurements (as in Zardini et al., 2006). The measured change in radius over multiple days was consistent with the model-derived radius (Fig. B1).



# 3 Results and discussion

## 3.1 Representative mass spectrum

Using the EDB-MS system, we obtained the mass spectrum of single particles that were trapped inside the EDB and then transmitted to the ionization source for vaporization and ionization. A sample mass spectrum of a PEG-200 particle (Fig. 3)

shows that the signal from the droplet can be easily detected.

## 3.2 Model-measure comparison

The measurement and model were compared for droplets of three different compositions: two binary mixtures and one more complex mixture.

### 3.2.1 Binary particles

Evaporation of both PEG-3 and PEG-4 were tracked in binary mixtures in which the second component was PEG-6, as an internal standard, with an initial molar ratio of approximately unity. The evaporation time extended to 60 minutes for the PEG-3 binary mixture and 170 minutes for the PEG-4 binary mixture. The spectra of 40 PEG-3 + PEG-6 droplets were collected in total. After filtering out particle with too small of a PEG-6 signal for reliable normalization (defined as less than 1000 counts s$^{-1}$, 20 particles) a total of 20 PEG-3 + PEG-6 particles remained for analysis. For the PEG-4 + PEG-6 particles,

the spectra of 15 particles were collected and all 15 had sufficient PEG-6 signal for quantification. Sizing information was only collected for 2 of the PEG-4 + PEG-6 particles and the bracketing model runs were necessarily defined by the two measured diameters. In each case, individual observations were binned into appropriate time intervals (10 and 20 minutes for PEG-3 and PEG-4, respectively). Due to the scatter in the data, the values in each bin were averaged and when multiple values were present in a bin, a bootstrap analysis was performed to estimate the uncertainty in the averaged value. The

results are compared to the predictions of the evaporation model in Fig. 4. The model was initialized with the known composition of the prepared binary mixtures.

After averaging over multiple droplets within each time bin the measured evaporation is consistent with the model rates for both PEG-3 and PEG-4, within considered uncertainties. As an alternate approximate check of the reasonableness of the relative measured evaporation rates that does not rely on the correctness of the model implementation, the approximate

evaporation timescales for PEG-3 and PEG-4 can be compared. Presuming all other conditions are held constant (starting radius, temperature, etc.) and temporarily neglecting the minor deviation from first-order decay due to the changing particle radius, the ratio of evaporation half-lives for PEG-3 and PEG-4 should equal the ratio of their vapor pressures, inverted. As shown in Fig. 4, the half-life for PEG-3 evaporation is about four times shorter than for PEG-4 (15 min versus 60 min), which is consistent with the PEG-3 vapor pressure being approximately four times larger than PEG-4 near 298 K (Table B1).





### 3.2.2 PEG-200 particles

Similar to the binary mixtures, the evaporation of PEG-200 particle components was also tracked. Following the same filtering procedure as for the binary particles, spectra were collected for 90 particles, and after filtering out particles with insufficient PEG-6 signal (63 particles), 27 particles remained for analysis. The same binning, averaging and boostrapped uncertainty procedure was performed as for the binary particles (with 10, 20, 400 and 400 minute bins for PEG-3, PEG-4, PEG-5 and PEG-7, respectively). The major components of the stock solution used to prepare these particles consisted of PEG-3 through PEG-7; the model-measurement comparison for each molecule, using PEG-6 as an internal standard, is given in Fig. 5. Here, because the starting composition of the purchased PEG-200 mixture was not available, the model was initialized with the average PEG composition given by the measurements of particles that were immediately ejected from the EDB. Again, the measured change in composition with time is largely consistent with modelled evaporation. The slight increase in PEG-7 over the longest model timescales is due to the faster evaporation of PEG-6 compared to PEG-7.

### 3.3 Extracting vapor pressures from model fit

The above analysis uses a model to check the appropriateness of the observed evaporation rates. However, one future utility of such an experimental system may be calculating the vapor pressure (or activity) of a compound for which the value is not known. Thus, we also assessed how well we can constrain the vapor pressures of compounds by optimizing the model fit to the experimental measurements. For both the PEG-3 + PEG-6 and PEG-4 + PEG-6 binary mixtures, we iteratively ran the model with the PEG-3 or PEG-4 reference vapor pressure (at 298.15 K) as the free variable (assuming PEG-6 to represent a reference compound with well-constrained vapor pressure). We performed the analysis twice for each mixture, fixing the temperature and initial diameter to each of the bracketing cases described above. We calculated the root-mean-squared deviation (RMSD) of the binned data points from the model output at the bin's midpoint time, and searched for convergence to a minimum RMSD. In each case, we found model convergence to the binned data. The extracted vapor pressures of PEG-3 and PEG-4 averaged over the two bounding temperature/diameter cases (48±12 mPa for PEG-3 and 14.6±2.2 mPa for PEG-4) are consistent with the literature vapor pressures ($66.8^{+11.0}_{-9.5}$ and $16.9^{+1.1}_{-1.0}$ mPa for PEG-3 and PEG-4, respectively), when their respective uncertainties are considered. The results demonstrate that the current data set allows calculating vapor pressures with uncertainty within a factor of 2. Because vapor pressure values derived from different experimental techniques can vary by orders of magnitude, even the precision obtained in this proof-of-concept measurement can represent a helpful constraint for compounds less well-studied than PEG (Bilde et al., 2015). The variability in the starting diameter and temperature are the dominant sources of uncertainty in this model fit, so the precision of extracted vapor pressures is expected to improve with better constraints on the particle-to-particle variability in EDB temperature and starting diameter.



### 3.4 Accounting for particle-to-particle signal variability

We have shown the experimental results to be consistent with expectations reflected in a kinetic model, and further, that meaningful vapor pressure values can be extracted if the values are assumed to be unknown. However, it can also be seen from Fig. 4 and Fig. 5 that there is considerable particle-to-particle variability in the signal for replicates collected after the

same EDB residence time. Additionally, this variability appears to differ between evaporation data sets: the PEG-4 + PEG-6 binary particle data appears much more tightly clustered than the PEG-3 + PEG-6 data set, for instance, meaning averaging over fewer points is required. This variability highlights the importance of averaging over multiple particles to obtain a quantitative picture of the change in droplet composition. We investigated possible sources of this variability in order to understand possible sources of improvement for future iterations of this system.

Because this variability is observed for particles for which little evaporation has occurred it seems unlikely that variability in the rate of evaporation is the cause. Instead, the variability more likely originates from vaporization, ionization, or the mass spectral measurement itself.

We investigated a number of possible factors contributing to the variability in signal. The variability was not explained by the variability in measured starting particle diameter. We compared the variability in signal for particles that were trapped in

the EDB to particles that were allowed to travel directly to the ionization region, either by passing through the EDB without first being charged, or by ejection out of the droplet generator positioned directly above the top of the transfer tube. Comparing the two sets of measurements on a solution-by-solution basis, we did not find that the particles first trapped in the EDB systematically demonstrated a larger variability in signal. This implies the trapping process is not a dominant source of variability in the signal.

Because there was particle-to-particle variability in the total amount of particle-derived signal measured by the MS, we looked for a correlation between normalized signal variability and the total raw counts, both for particles trapped in the EDB and immediately ejected, and for particles that were never trapped. Our data set was too sparse to make statistically rigorous conclusions, but it did not appear as if there was a consistent relationship for all studied solutions between total raw counts and the normalized signal variability, once we filtered out data with raw PEG-6 peak intensities judged too low (<1000

counts s$^{-1}$) to allow for accurate peak height determination.

We also considered the possibility that the 1 s MS sampling time used for these experiments could be undersampling the pulse of a signal from the vaporized particle, which could lead to added signal variability. To check this, we compared the variability in normalized peak signals of PEG-200 particles when the MS sampling interval was 1 s or 0.1 s. In each case, the PEG-200 particles were injected into the EDB and then immediately transferred to the ionization region, without trapping.

We found no difference in the variability in the normalized PEG-3 through PEG-7 signals between the 1 s and 0.1 s sampling data.

Another factor to consider is the possibility of variability in the vaporization process. The quantification procedure presumes the vaporization of analyte molecules is virtually instantaneous, for every particle measured. If the vaporization process were





not instantaneous for certain lower-volatility analyte molecules, this would manifest itself as the signal intensity being spread out over a longer time interval, with diminished peak intensity. The peak signals for most particles showed a consistent sharp peak shape on all analyte signal channels. In a small number of cases, it was observed that heavier molecular weight (i.e., lower volatility) PEG showed an anomalously broad distribution of signal in time, with a smaller peak intensity, whereas lighter molecular weight PEG showed the normal sharp peak. These cases were ascribed to irregular vaporization, perhaps due to misalignment of the particle transfer, and discarded from further analysis. Had these data not been discarded, they would have contributed extremely large "normalized signals" for lighter molecular weight PEG, since the PEG-6 peak intensities were weakened due to broadening. If irregular vaporization contributed to the variability in normalized signal observed in Fig. 4 or Fig. 5, it would have needed to have arisen from cases in which the broadening of the lower volatility signals was too subtle to be screened out by eye.

From these checks, we observed what appeared to be an inherent variability of approximately ±20–30% in the normalized peak intensities, regardless of the raw signal strength, initial particle diameter, the MS sampling rate, or whether or not the particle was held in the EDB prior to transfer to the ionization region. The origin of the apparent somewhat greater variability for some data, such as the first time bin for the PEG-3 + PEG-6 binary particles, has not been determined. Future work is needed to determine whether such variability persists in future studies with refinements to the experimental design.

An additional limiting factor for this experiment, beyond the signal variability and consequent need for averaging, was the difficulty of transfer from the EDB to the vaporization region for some particles. It was found that the transfer protocol described above worked with near-100% success for transferring particles that were ejected from the EDB relatively quickly after their initial formation. However, the transfer success rate for some particles was found to become appreciably lower when residence times in the EDB extended to hours or days. We hypothesize this is a result of the smaller remaining particle mass after a significant portion of the particle's starting material has evaporated: the less massive particle is then more buffeted by any turbulence it encounters during the transfer step and is less likely to strike the vaporization platform. Future experiments with this system would be aided by an improved transfer design in which lighter particles also reach the ionization source with near-100% efficiency, or an experimental design in which the final droplet mass is not much smaller than the initial.

## 4 Conclusions

In this work, we describe a new electrodynamic balance–mass spectrometer system that is capable of suspending a single particle of known starting composition within a bath gas of controlled composition for an arbitrarily long amount of time, and then measuring the particle's composition via mass spectrometry after transfer to an ionization source. We demonstrate the ability of the EDB-MS system to assemble a series of snapshots tracking how a particle's chemical composition changes with time, here with a model system of polyethylene glycol components whose composition changes due to evaporation.



Because evaporation of polyethylene glycols has been carefully studied, we are able to validate our experimental results by means of a comparison to a simple kinetic model of evaporation.

For single-component aerosol particles, existing EDB-based techniques to measure vapor pressure by continuously monitoring the change in diameter currently offer more precision, due to the high accuracy with which the diameter can be
measured compared to the larger variability in mass spectrum-derived peak ratios. However, since the mass-to-charge ratio of the quantified mass spectral peak provides information about the chemical identity of the compound whose evaporation is tracked, the EDB-MS approach is less vulnerable than a diameter-tracking method to measuring incorrect evaporation rates due to the presence of impurities. Furthermore, the EDB-MS approach can be applied to a wider range of systems with greater chemical complexity due to the inherently multichannel detection technique, as demonstrated by the PEG-200
evaporation experiment. Additional systems that may lend themselves to study by the EDB-MS are discussed below.

Further improvements to the experimental set-up are possible, beyond the prototype design used for this experiment. The EDB can be altered to improve the gas-flow control and measurement, and make possible monitoring particle sizing by means of light scattering (e.g., as in Zardini et al., 2006). The design of the particle transfer from the EDB can be improved to increase the transfer efficiency for the smallest-diameter particles, which are most sensitive to turbulence in gas flow.
Implementing an alternate ionization scheme could remove the limitation of only detecting molecules that are sufficiently volatile to vaporize quickly upon impact on the 220 °C platform. One approach may be to vaporize the particle not with a heated platform, but with an infrared laser, possibly followed by a separate gas-phase ion source (as in Warschat et al., 2015; Westphall et al., 2008; or Zelenyuk et al., 2009). Alternately, an electrospray-type scheme may be used, such as delivering the particle onto a paper spray source (as in Jacobs et al., 2017), ionizing via an interaction between the particle and a spray
of ions (e.g., as in Doezema et al., 2012; Gallimore and Kalberer, 2013; or Horan et al., 2012), or by producing a spray directly from the particle when dropped onto a charged needle tip (as in Tracey et al., 2014).

## 4.1 Future experiments

We envision future experiments using this EDB-MS system to study linked chemical and physical transformations of particles, particularly with relevance to atmospheric aerosol particles. The strength of this system lies in its ability to couple
the strengths of trapped single particle experiments—in which a single particle transforms over a timescale of minutes, hours, or days, with careful control of both condensed- and gas-phase compositions—with the chemical specificity of mass spectrometric analysis. These strengths make the EDB-MS a complimentary technique to existing experimental and modeling approaches.

One class of future experiments is to measure evaporation rates. By fitting to a model, evaporation data can be used to
determine vapor pressures of compounds when the levitated particle represents an ideal mixture, as was demonstrated in the current work. In addition, particles can be prepared for which evaporation is not expected to proceed as for an ideal mixture. For instance, the compound being studied may be expected to have an activity coefficient in the particle mixture deviating significantly from unity, or evaporation might be kinetically limited due to physical properties (slow diffusion or phase



separation). If the vapor pressure of the compound under study is known, the observed evaporation rate can be compared quantitatively to a model representation of that nonideality. The use of a mass spectrometer as the analytical technique in these cases means the presence of all MS-detectable components of the particle can be tracked with time, providing a more detailed data set than an equivalent experiment measuring particle diameter alone.

Beyond measurements of evaporation in ideally and nonideally mixed particles, the EDB-MS can be used to track chemical reactions in particles. A condensed phase mixture that itself is reactive can be prepared and injected as a particle, or the composition of the bath gas be changed to induce changes in the particle's composition. One example would be to change the RH of the gas, which can change the water activity in the particle and consequently affect condensed-phase chemistry by either serving as a plasticizer to help speed diffusion in a kinetically "frozen" particle, or affecting equilibrium of a chemical

reaction in which water directly plays a role, such as hydrolysis. A second example would be to add an oxidant such as ozone or the hydroxyl radical (OH) to the gas, which can cause organic molecules in the particle to undergo rounds of oxidative "aging".

In all of these examples, the EDB-MS is well-suited to studying chemical transformations over "long" timescales of hours or days, which are of interest because of their relevance to the atmospheric lifetime of aerosol particles. One approach in

laboratory studies of aerosol chemistry has been to speed up the reaction of interest by increasing the concentration of a reactive species, for example, ozone or OH, compared to typical atmospheric concentrations. The technique presumes the chemical changes an aerosol particle undergoes over multiple days can be accurately compressed to a shorter timescale by working at higher concentrations. However, it may be the case that working at higher concentrations masks other processes that are important on a longer timescale, but do not speed up under the selected concentration conditions. Unimolecular

reactions are an example of a class of such processes. Working with the EDB-MS would provide an opportunity to check whether the results from high concentration experiments can in fact be extrapolated to slower, lower-concentration conditions in the atmosphere.

This system therefore represents a valuable analytical tool for better understanding fundamental physicochemical processes of aerosol particles, whose value in part lies in providing improved model representations of these processes, enabling a

better understanding of the role of aerosol particles in human health and climate.

**Data and code availability**

The data set containing mass spectrum peak intensities and sizing data for the full set of particles measured in this study is available upon request. The particle evaporation model code is available at https://github.com/awbirdsall/pyvap.

**Appendix A: Characterization of mass spectrometer fragmentation and sensitivity**

The appendix includes data on PEG mass spectral relative sensitivities (Table A1) and fragmentation patterns (Table A2).





**Appendix B: Kinetic model of particle evaporation: parameters used and check of model performance**

The appendix includes a table of parameters for PEG used in the kinetic evaporation model (Table B1) and a figure illustrating a check of the performance of that model (Fig. B1).

**Author contributions**

5   UK and FK initially conceived of the work. AB and UK developed the laboratory setup, performed the experiments, and analyzed the data. AB developed the evaporation model. AB prepared the manuscript with contributions from UK and FK.

**Competing interests**

The authors declare they have no conflict of interest.

**Acknowledgments**

10   This material is based upon work supported by National Science Foundation grant CHE 1213723, as well as the National Science Foundation Graduate Research Fellowship under grant numbers DGE 1144152 and DGE 1745303. The authors thank Michael Greenberg and Uwe Weers for engineering assistance, Adam Trevitt for helpful preliminary discussions, and Kevin Wilson for sharing an early draft of a manuscript.





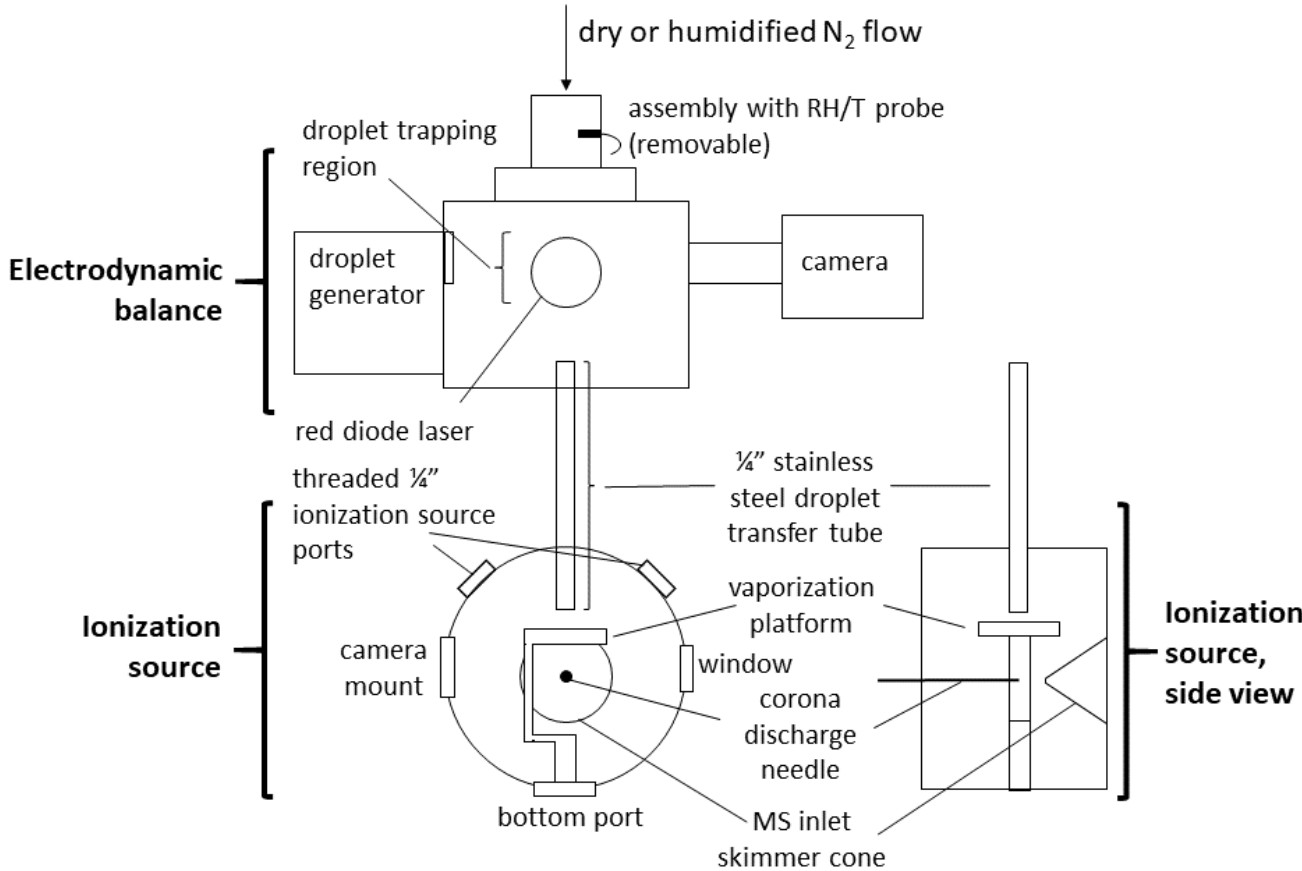

**Figure 1:** Schematic of experimental setup. A ~140 pL droplet is ejected from the inkjet cartridge, charged, and trapped in the electrodynamic balance (EDB). Once the droplet is ready to be destructively analyzed by the mass spectrometer, it is transferred out of the EDB, down the transfer tube, and to the ionization source. In the ionization source, the droplet strikes the heated vaporized platform (220 °C) and the resulting analyte vapors are drawn toward the mass spectrometer (MS) inlet. The corona discharge from a high-voltage needle ionizes the analyte molecules (positive mode) before the molecules enter the MS. The transfer tube terminates ~4 mm above the vaporization platform. The tip of the corona discharge needle is ~2 cm in front of the MS inlet skimmer cone.





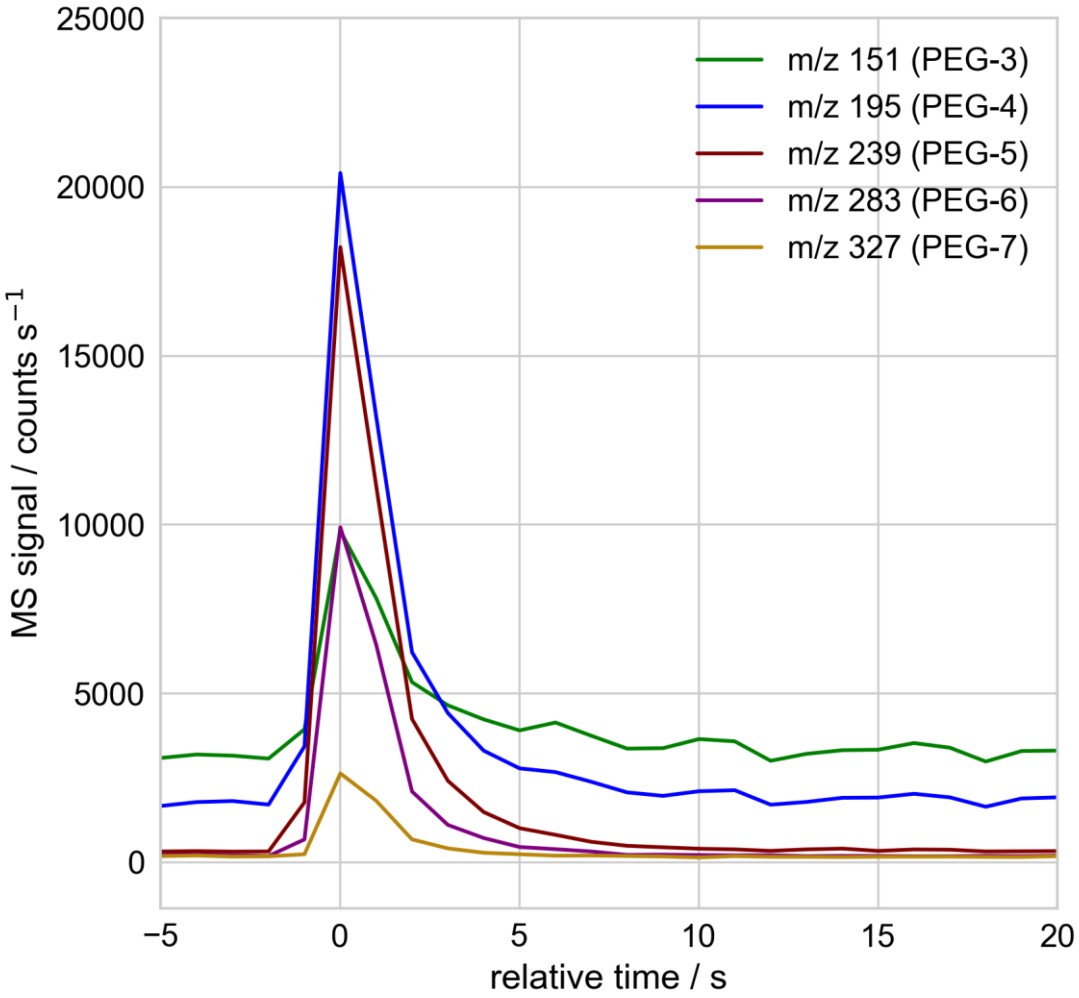

**Figure 2:** Sample mass spectrometer selected ion time series used to quantify the droplet's molecular components, with 1 Hz sampling. Here the time series corresponding to the mass spectrum of Fig. 3, of a single PEG-200 droplet, is shown. The signal intensity in each mass channel is recorded as the peak height above surrounding background, using a peak-detection algorithm checked by eye for correctness. The relative abundance of each PEG molecule is then obtained after correcting for the empirically-determined relative signal response of each PEG molecule (Table A1).





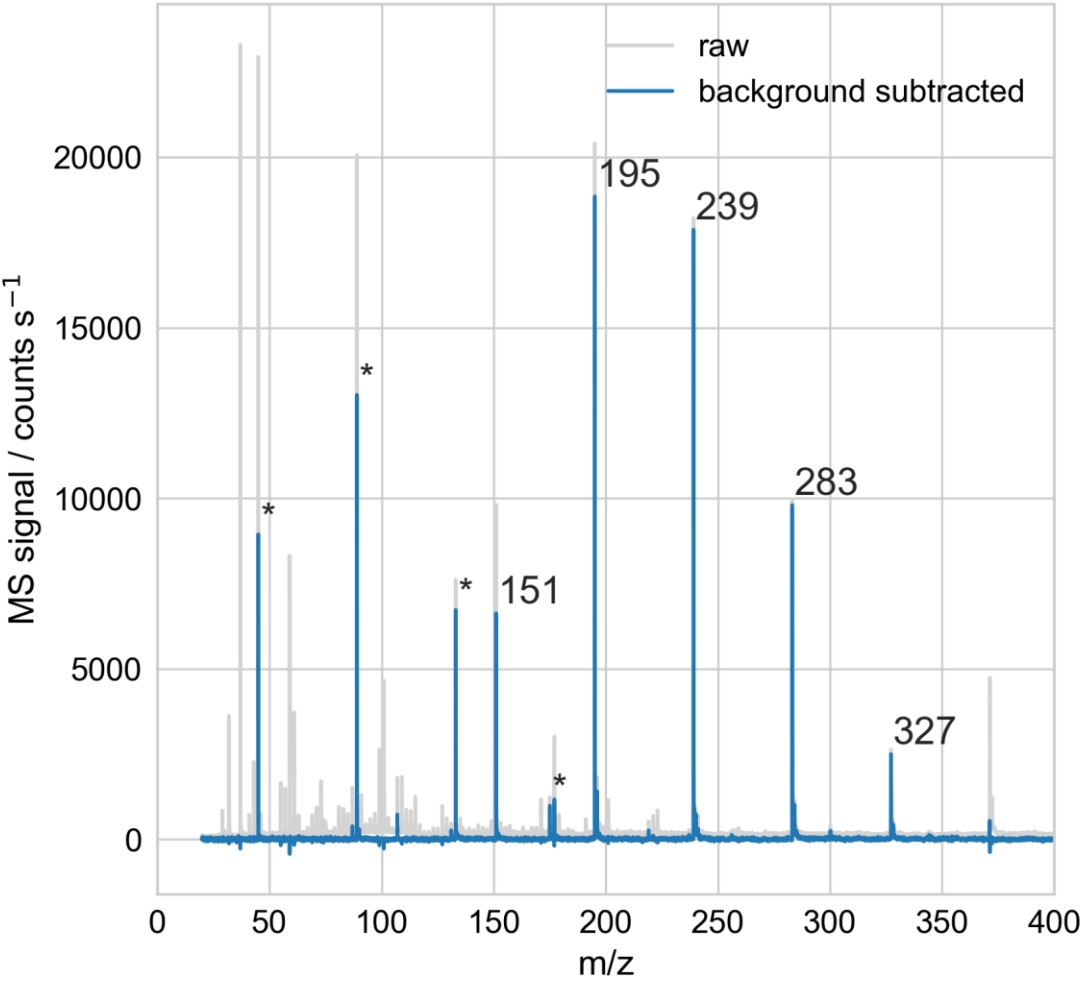

**Figure 3:** Sample mass spectrum of a droplet consisting of polyethylene glycol, average molar mass 200 g mol⁻¹ (PEG-200). The droplet was trapped in the electrodynamic balance and then transferred to the ionization source for analysis, as in Fig. 1. The peaks at 151, …, 327 m/z, with regular 44 m/z spacing, correspond to MH⁺ for M = triethylene glycol (PEG-3) through heptaethylene glycol (PEG-7). Peaks at 45, 89, 133, and 175 m/z (marked with *) arise in part from PEG fragmentation, confirmed with mass spectral analysis of single-component PEG droplets. All other peaks originate from the mass spectrum background, which reflects air drawn into the MS from both the laboratory and the EDB-MS assembly.





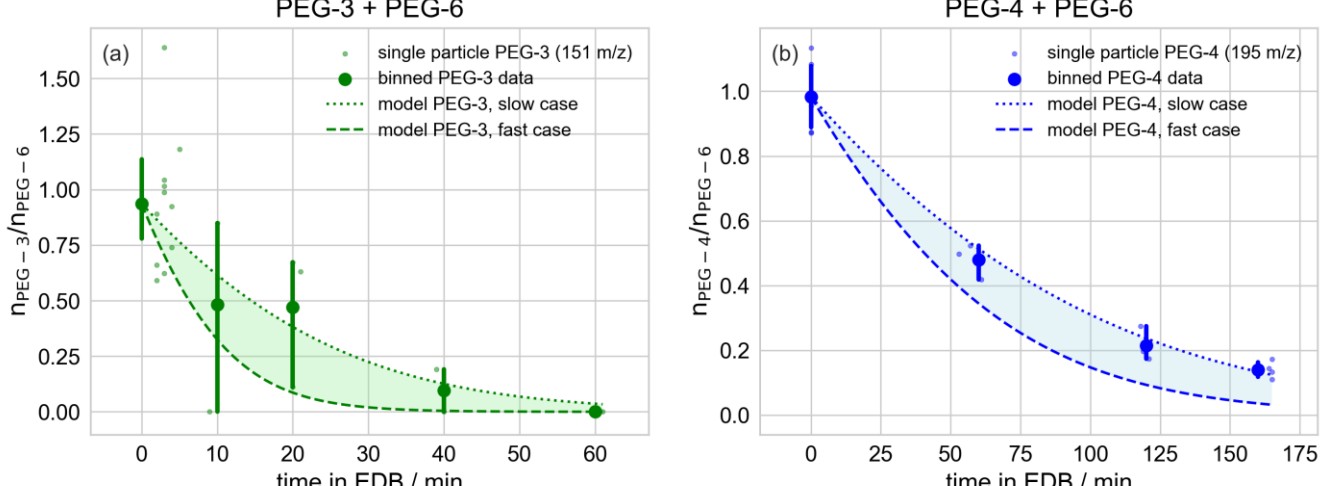

**Figure 4:** Evaporation of polyethylene glycol (PEG) droplets of binary composition, with starting molar ratio approximately 1:1. Initial droplet compositions are (a) triethylene glycol (PEG-3) and PEG-6; (b) tetraethylene glycol (PEG-4) and PEG-6. All values are molar ratios, scaled to in-droplet hexaethylene glycol (PEG-6) abundance as an internal standard. Experimental observations are binned by time (10 and 20 minute intervals for PEG-3 and PEG-4, respectively) and the mean value is plotted as a point. When multiple data are available within a single bin, a 95% confidence interval is estimated via a bootstrap analysis and plotted. Outputs from kinetic models of PEG evaporation are plotted as shaded regions. The regions are bounded by limiting cases reflecting the variability in the EDB air temperature and the droplet starting diameters, and the uncertainty in literature vavpor pressures (upper curve: lowest temperature, largest droplet, and lowest vapor pressure; lower curve: highest temperature, smallest droplet, and highest vapor pressure).

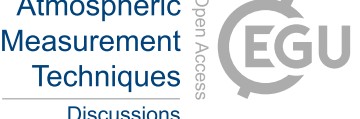

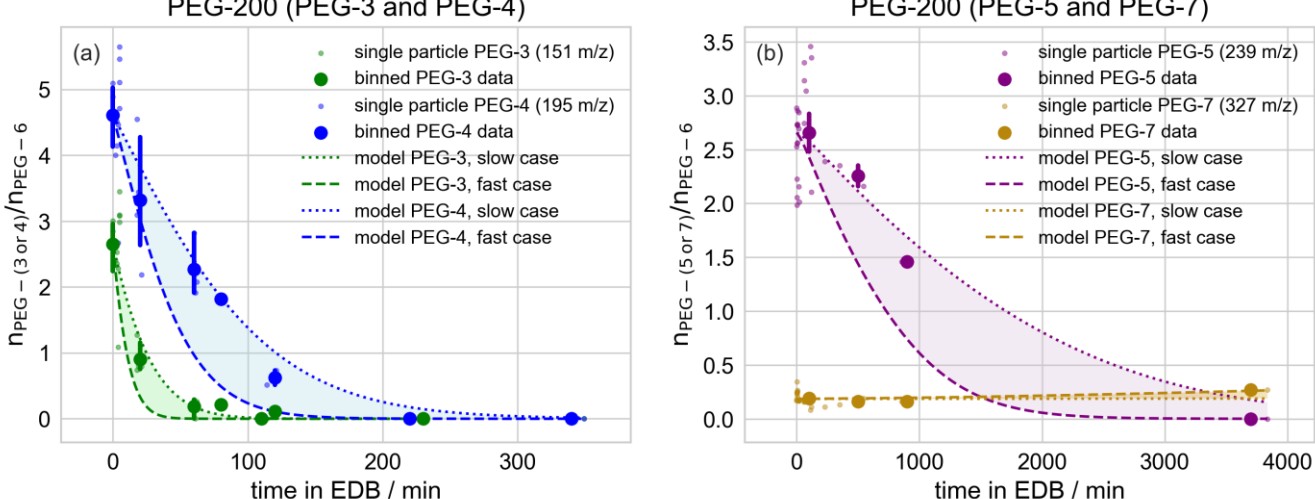

**Figure 5:** Evaporation of PEG-200 droplets of mixed composition, as in Fig. 4. Here, observations are binned in 10, 20, 50, and 50 min intervals for PEG-3, PEG-4, PEG-5, and PEG-7, respectively. The same data set is used for tracking the evaporation of all PEG molecules; plots are split across two figures ((a): PEG-3 and PEG-4; (b): PEG-5 and PEG-7) due to the significantly differing evaporation timescales.



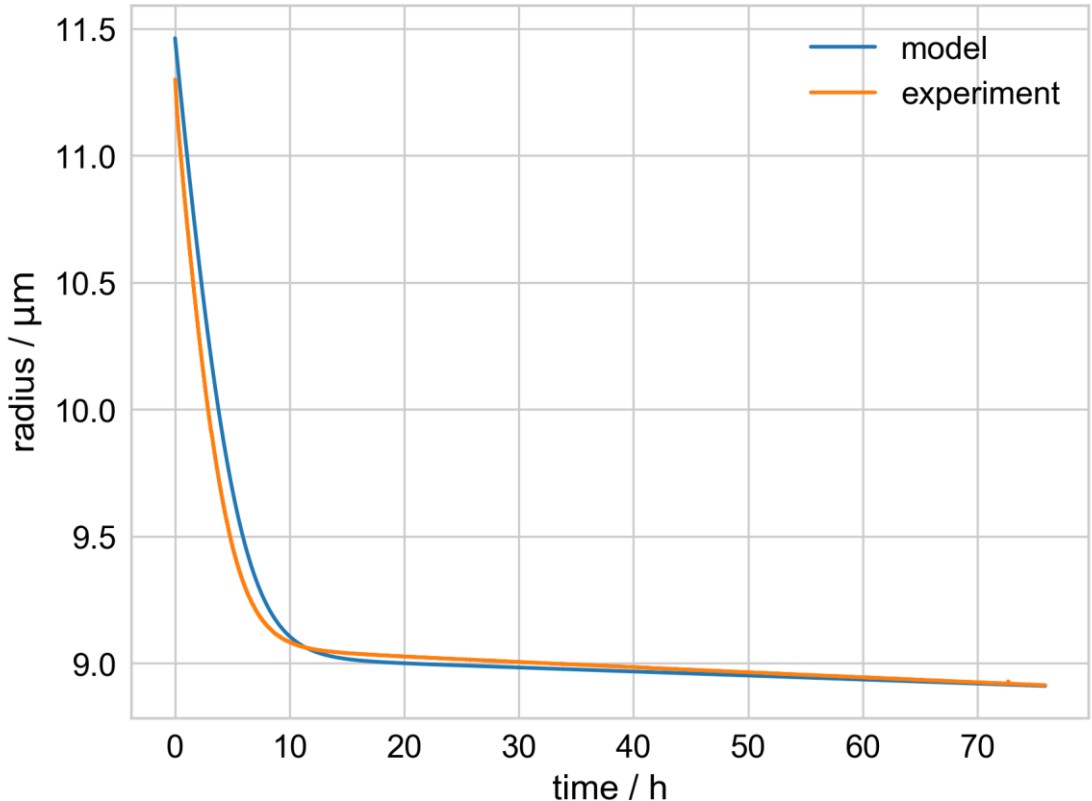

**Figure B1:** Comparison of ETH evaporation experiment and model, used to check model performance. A droplet generated from a solution of known PEG-4 and PEG-6 composition, along with water, was injected into an electrodynamic balance (different from the one used during the EDB-MS experiments). The droplet's radius was monitored continuously while levitated in the EDB by fitting the scattering spectrum of incident light (Zardini et al., 2006). The kinetic model of evaporation was initialized with the source solution's molar ratio of PEG-4 and PEG-6. The modeled radius, derived from the modeled molecular composition as evaporation takes place, is compared to the experimentally determined radius. Because it took several minutes for the conditions in the EDB to stabilize following injection, the model was initialized not with the experimentally measured starting radius but with a starting radius such that the model radius agreed with the experimentally measured radius at the final time (approx. 75.9 hours). Once equilibrated, the ambient relative humidity (i.e., water activity) in the EDB was measured to be 12%. A fixed water activity of 0.12 was estimated to correspond to a mole fraction of water of 0.18 in the particle, from a previous experimental study of water–PEG-200 mixtures (Ninni et al., 1999), and this fixed mole fraction of water was included in the model in addition to the PEG-4 and PEG-6. The model was run with the mean experimentally measured temperature in the EDB, 291.06 K.



**Table A1:** Relative sensitivity of the EDB-MS system to PEG-3 through PEG-6, normalized to PEG-6. The relative sensitivity is defined as value by which the molar ratio of a particle's composition (PEG-*X*/PEG-6, $X$ = 3, 4, or 5) is multiplied to obtain the ratio of peak intensities measured by the MS. The values were obtained by averaging the measured peak ratios of binary particles, consisting of PEG-6 and one of PEG-3, PEG-4 or PEG-5, that were injected into the EDB, trapped momentarily, and then immediately transferred to the ionization region for measurement. The relative sensitivity of PEG-7 was not measured and was assumed to equal 1.

|  | PEG-3 (m/z 151) | PEG-4 (m/z 195) | PEG-5 (m/z 239) | PEG-6 (m/z 283) |
|---|---|---|---|---|
| Rel. sens. | 0.33 | 0.72 | 0.94 | 1.00 |



**Table A2:** Fragmentation patterns of individual PEG molecules. Peak intensities are normalized to the MH[+] signal for each molecule. For each molecule, intensities are the mean (with 1-σ standard deviation) of 10 background-subtracted spectra of particles that are injected into the EDB and immediately travel to the ionization source, without being trapped in the EDB for any amount of time. Only m/z values with intensity of at least 5% of the parent ion for at least one PEG molecule are listed.

| m/z | 45 | 87 | 89 | 133 | 151 | 175 | 195 | 239 | 283 |
|-----|-----|-----|-----|-----|-----|-----|-----|-----|-----|
| PEG-3 | 89±5 | 13±1 | 100±5 | 12±1 | 100 | - | - | - | - |
| PEG-4 | 17±1 | 2±3 | 28±2 | 17±1 | 1±1 | 6±2 | 100 | - | - |
| PEG-5 | 7±1 | - | 12±0 | 12±0 | 1±2 | - | - | 100 | |
| PEG-6 | 5±2 | - | 7±0 | 7±0 | - | 8±1 | - | - | 100 |



**Table B1:** Properties of PEG molecules used in the particle evaporation model: $D_g$ is gas-phase diffusivity at 298 K and 1 atm, M is molar mass, ρ is density at 298 K, $P^0_{vap}$ is saturation vapor pressure at a reference temperature of 298.15 K, $\Delta H_{vap}$ is enthalpy of vaporization. The model disregards the temperature dependence of any value, with the exception of vapor pressures being calculated from $P^0_{vap}$ and $\Delta H_{vap}$ using the Clausius-Clayperon equation. Values are taken from Krieger et al. (2017), in which vapor pressure measurements represent consensus values from a study by multiple research groups using different setups for detecting vapor pressures over a large temperature range.

| | $D_g$ / $10^{-6}$ m² s⁻¹ | M / g mol⁻¹ | ρ / g cm⁻¹ | $P^0_{vap}$ / Pa | $\Delta H_{vap}$ / kJ mol⁻¹ |
|---|---|---|---|---|---|
| PEG-3 | 5.95 | 150.2 | 1.108 | $6.68^{+1.10}_{-0.95} \times 10^{-2}$ | 78.3±0.7 |
| PEG-4 | 5.20 | 194.2 | 1.132 | $1.69^{+0.11}_{-0.10} \times 10^{-2}$ | 77.1±0.4 |
| PEG-5 | 4.66 | 238.4 | 1.155 | $5.29^{+0.75}_{-0.65} \times 10^{-4}$ | 90.6±1.1 |
| PEG-6 | 4.26 | 282.3 | 1.180 | $3.05^{+0.59}_{-0.49} \times 10^{-5}$ | 102.1±1.5 |
| PEG-7 | 3.94 | 326.4 | 1.206 | $1.29^{+0.48}_{-0.35} \times 10^{-6}$ | 113.7±2.7 |





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
