# Peer review of "Electrodynamic balance–mass spectrometry of single particles as a new platform for atmospheric chemistry research"

_Atmospheric Measurement Techniques, 2017_

## Referee Comment (RC1) · Anonymous Referee #1 · 9 Oct 2017

General Comments:

This manuscript describes an instrument that allows individual particles several micrometers in diameter to be trapped in an electrodynamic balance (EDB) over a period of hours to days where various physical and/or chemical transformations may take place, followed by online chemical analysis with mass spectrometry. In the current version of the instrument, particles exiting the EDB strike a heated probe, where molecular species are vaporized and then ionized in a corona discharge. This combination of techniques represents a significant addition to the arsenal of trapping methods used to study physicochemical processes relevant to atmospheric aerosol. The authors do

a nice job of reviewing previous work with EDB technology and/or mass spectrometric detection. The authors might consider including a sentence or two on the use of optical traps since these are also used with great success to study atmospherically relevant particles.

Specific Comments:

1. The topic of droplet size and the ion signal intensity produced from it should be discussed more coherently in the manuscript. The authors mention 10-30 micrometers in the introduction, but it is not clear whether this is limited by the balance or mass spectrometer. In section 2.4, the authors indicate an initial droplet volume for their experiment of ~140 pL, which then shrinks as water evaporates. In section 2.6 they assume a droplet size of 9-11 um in their model. However, droplet size and/or analyte mass sampled is not given in the discussion or captions of Figures 2 and 3. Nor is it discussed in any detail Section 3.4, which otherwise gives an excellent discussion of particle-to-particle signal variability. One gets the sense that the absolute signal variability is much greater than the 20-30% normalized signal variation reported by the authors. Furthermore, it seems that the PEG droplet size studied in this experiment represents a practical lower limit, since working with a smaller size would probably have led to an unacceptably high fraction of particle spectra being filtered out from further analysis. This is not a criticism of the current work, since the method is very powerful in its current incarnation. However, the topic could use more attention in the text.

2. Section 2.2. Spring point measurements were made with 18 um diameter PMMA spheres. Is it possible that the existence of doublets and/or larger aggregates of PMMA spheres could have influenced the results?

3. Section 3.2.1. "Particle" in line 13 of this page (fourth line of the section) should be plural.

---

## Referee Comment (RC2) · Anonymous Referee #2 · 13 Oct 2017

This paper reports the coupling of an EDB microdroplet balance to a commercial mass spectrometer facilitated by a vaporization platform and corona-discharge ionisation source.

Droplets containing polyethylene glycol (PEG) - with varying molecular-weight ratios are trapped, and left to evaporate, and then the molecular composition of the droplet is analysed using MS. Relative ion abundances, normalised to PEG6, are plotted to model the evaporation of the droplet with molecule-specific detail ie. what proportions of species are left behind comprising the droplet.

The MS result generally fit well to an evaporation model. It is a very interesting new

technique and I think the first time we have seen these evaporation models applied directly to molecular-specific measurements (like MS). Since single droplet mass spectra are measured - the should be great prospects to almost completely control all the droplet properties (composition, size, charge etc) and the environment related to the measurement. There is no ensemble averaging.

There are a few papers emerging recently that are interfacing microdroplet sources with online mass spectrometry techniques - this is, in part, due to the robustness of commercial mass spectrometers that have atmospheric sources like ESI. So interfacing "ambient pressure" droplet sources to these mass spectrometers is rather straightforward as the instruments already come designed with the differentially-pumped to get from droplet to ion detection. Stable sampling is the technical challenge.

Other groups have reported using a paper-spray electrospray ionization (Jacobs et al.) and HV needle desorption (Tracey et al.) , I can also remember seeing some results using DART and single droplet analysis. I hope we will soon see some comparisons between these ionization techniques. Molecular dependent ionization and detection efficiency will always be a key issue.

This paper does well-discuss and highlight the issues around quantitative MS sampling of droplets - particularly when "single droplet" mass spectra are distinguished.

For single microdroplets, droplet-to-droplet variations affect desorption and ionization efficiency and - more insidiously - can affect ions differently. This is the well-known challenge with just about all forms of quantitative mass spectrometry. This paper discusses these concerns towards the end of the results section very well.

This is an excellent paper, a great read and I recommend publication. I dont have any further specific comments or corrections to add.

---

## Referee Comment (RC3) · Anonymous Referee #3 · 25 Oct 2017

The manuscript demonstrates how the coupling of two well-established techniques - single-particle mass spectrometry and electrodynamic particle trapping - results in a new approach significantly advancing the capabilities of the individual parts. The manuscript is very well and clearly written and can be published without modifications. I have actually no major comments, just two remarks below:

1. A combination of electrodynamic trap and mass spectrometer has been described previously by (Dale et al., 1994), who used laser ablation to ionize the trapped particle. Please give a mention of this work in your review of analytical techniques.

2. The possibility of studying the slow gas-phase chemical reactions in atmospheric

aerosols is given as one of the motivations for this work. Aerosol particles, however, are much smaller in size (typically submicron) than the particles used in your work, so that trapping a particle in the EDB and transferring it into the ionization section might be very difficult. Could you discuss this issue in the outlook section? As a layman's suggestion, wouldn't it be possible to use a linear quadrupole trap to move a train of particles slowly through the reaction zone delivering them one by one into the ablation section coupled to the inlet of a mass spec? Such technique, though applied for optical particle characterization and not for mass spectrometry, has been recently described by (Sivaprakasam et al., 2017).

References:

Dale, J. M., Yang, M., Whitten, W. B. and Ramsey, J. M.: Chemical Characterization of Single Particles by Laser Ablation/Desorption in a Quadrupole Ion Trap Mass Spectrometer, Anal. Chem., 66(20), 3431–3435, doi:10.1021/ac00092a021, 1994.

Sivaprakasam, V., Hart, M. and Eversole, J. D.: Surface Enhanced Raman Spectroscopy of Individual Aerosol Particles, J. Phys. Chem. C, 121, 22326−22334, doi:10.1021/acs.jpcc.7b05310, 2017.

---

## Author Comment (AC1) · 15 Nov 2017

**Author Response to Anonymous Referee #1 of "Electrodynamic balance-mass spectrometry of single particles as a new platform for atmospheric chemistry research" by A.W. Birdsall et al.**

We thank the referee for their thoughtful comments, which have helped improve the manuscript. Our replies are below (referee comment in **bold**, response in normal face, new manuscript content in *italics*, removed manuscript content in <del>strike-through</del>).

**The authors might consider including a sentence or two on the use of optical traps since these are also used with great success to study atmospherically relevant particles.**

We agree that optical traps are a highly valuable platform for single particle studies. The introduction's treatment of levitated droplet experiments does acknowledge that optical, electrodynamic, and acoustic traps have all been used to study levitated particles. To clarify and emphasize the important contributions of both EDBs and optical traps, we will amend the Introduction as follows (p. 1, line 27):

"Researchers have studied particles deposited onto a substrate, or alternately, particles levitated by means of a "trapping" force originating from an electric field, radiation pressure of a laser beam, or acoustic wayes. Levitated droplet experiments are appealing because they mimic aerosol particles in the ambient environment in certain key ways: the presence of a surrounding bath gas, an enhanced surface-to-bulk ratio, the absence of physical contact with a substrate, and the ability to study supersaturated particles. Multiple laboratories have analyzed levitated droplets using optical techniques, which can be used to track evaporation and condensation via highly precise particle sizing, as well as some changes in chemical composition (reviewed in Krieger et al., 2012). Using electrodynamic or optical forces, multiple laboratories have analyzed levitated droplets using optical techniques such as Raman spectroscopy and Mie resonance spectroscopy (earlier work reviewed in Krieger et al., 2012). A number of different properties have been studied in this way, including vapor pressures (Cai et al., 2015; Cotterell et al., 2014; Huisman et al., 2013; Krieger et al., 2017), hygroscopic growth (Cai et al., 2015; Cotterell et al., 2014; Rovelli et al., 2016), optical properties (Mason et al., 2015), liquid-liquid phase separation (Stewart et al., 2015), diffusivities and diffusion coefficients (Bastelberger et al., 2017; Lienhard et al., 2014), and oxidative aging (Dennis-Smither et al., 2014)."

**The topic of droplet size and the ion signal intensity produced from it should be discussed more coherently in the manuscript.**

We thank the referee for a number of suggestions to clarify the manuscript on the topic of droplet size and ion signal intensity. We address the individual points below.

The authors mention 10-30 micrometers in the introduction, but it is not clear whether this is limited by the balance or mass spectrometer.

The EDB can operate with smaller diameters; the operating diameter was chosen for a number of other factors (particle injection, ease of transfer to ionization region, magnitude of MS signal). We will clarify discussion of size range in the Introduction with the following change (p. 2, line 28):

"Here we describe a newly-developed system that couples an electrodynamic balance (EDB), which levitates aerosol particles with diameters on the order of 10–30 micrometers for an arbitrarily long amount of time, with mass spectral analysis of the entire particle. We operate with particles of diameter approximately 10–30  $\mu$ m for reliable acquisition of quantifiable mass spectra, though the EDB can levitate particles of smaller diameter."

**[...] droplet size and/or analyte mass sampled is not given in the discussion or captions of Figures 2 and 3.**

We will clarify in the Figure 2 and 3 captions and surrounding body text that the figures are from a 20  $\mu m$  diameter particle:

§ 2.5 (p. 6, line 20): "Fig. 2 presents a sample time trace of selected ion signals arising from ejection and ionization of a 20  $\mu$ m diameter PEG-200 particle."

§ 3.1 (p. 8, line 4): "A sample mass spectrum of a PEG-200 particle (Fig. 3) shows that the signal from the droplet droplets with diameters on the order of 20  $\mu$ m can be easily detected."

Fig. 2 caption (p. 16, line 3): "Here the time series corresponding to a mass spectrum of Fig. 3, of a single PEG-200 droplet *with diameter 20 \mum*, is shown.

Fig. 3 caption (p. 17, line 2): "Sample mass spectrum of a droplet consisting of polyethylene glycol, average molar mass 200 g mol-1 (PEG-200), *droplet diameter 20 \mum*.

**Nor is it discussed in any detail Section 3.4, which otherwise gives an excellent discussion of particle-to-particle signal variability.**

For particles used in the evaporation experiments the variability in measured diameters, using the spring-point technique, was found not to explain the variability in normalized signal as evaporation took place. In other words, the particle-to-particle variability was dominated by factors other than particles with  $\sim 9 \mu m$  radius evaporating faster than particles with a  $\sim 10 \mu m$  radius. This is what was meant in the manuscript by the statement, "The variability was not explained by the variability in measured starting particle diameter." (§ 3.4, p. 10, line 13)

We have since performed additional checks on the role of size variation in ion signal intensity and variability:

• We considered PEG-200 particles which were trapped and "immediately" ejected from the EDB (within 6 minutes of introduction), that were measured with the spring-point technique, and met the threshold of having sufficient signal (defined as at least 1000 counts at the m/z 283, PEG-6 channel). For these particles, there was no correlation between the particle diameter and raw signal. There is insufficient data to determine

whether within this size range there is any systematic difference in the amount of particle-to-particle signal variability.

• We have other data consisting of mass spectra for PEG-200 particles generated from solutions of varying weight fractions of PEG-200 in water. The weight fraction of PEG-200 in water can be taken as proportional to the starting trapped particle mass after fast water evaporation, since particle diameters were not measured with the spring-point technique. We compared particles generated from 10 wt% and 25 wt% PEG-200 mixtures, again restricting to particles immediately ejected from the EDB and over the signal threshold (1000 counts at m/z 283). The particle-to-particle raw signal variability was less for droplets from the 25 wt% mixture, but the normalized signal variability was similar for droplets from the two mixtures.

To clarify and summarize this analysis, we will update § 3.4 and Appendix A, including a new figure and caption:

§ 3.4 (p. 10, line 13): "We investigated a number of possible factors contributing to the variability in signal. Within the size range of particles analyzed during the evaporation experiments, variability in the particle diameter (approx. ±10%), measured with the spring-point method, did not correlate to particle-to-particle variability in apparent evaporation rates, or to particle-to-particle variability in absolute signal. For two populations of PEG-200 particles with masses varying by a factor of approximately 2.5, higher variability in raw signal was observed for the smaller particles. In this data set the decreased particle-to-particle variability in normalized signal can also be readily observed (Fig. A1). From these analyses we conclude that though the particle-to-particle variability in raw signal may be affected by significant differences in particle mass, the The variability in the normalized evaporation data was not explained by the variability in measured starting particle diameter."

**Appendix A:**

p. 13, line 29: "Appendix A: Characterization of mass spectrometer fragmentation, <del>and</del> sensitivity, *and signal variability*"

p. 13, line 30: "The appendix includes data on PEG mass spectral relative sensitivities (Table A1) and fragmentation patterns (Table A2). *The role of normalization and particle size on particle-to-particle mass spectrum signal variability was analyzed (Fig. A1). Mass spectra for a set of PEG-200 particles were considered: those for particles "immediately" ejected from the EDB after trapping (defined as within 6 minutes of introduction), and with spectra over the signal threshold (defined as at least 1000 counts in the m/z 283, PEG-6 channel). The particles were generated from two different solutions of PEG-200 in water: 10% and 25% by weight. After fast water evaporation, the weight fraction can be taken as proportional to the starting trapped particle mass. (A full set of spring-point diameter measurements are unavailable.) The particle-to-particle raw signal variability (Fig. A1, top) was less for droplets from the 25 wt. % mixture, but the normalized signal variability (Fig. A1, bottom) was similar for droplets from the two mixtures. In both cases, the variability in the normalized signal is smaller than the variability in the raw signal."*

"Figure A1: Analysis of the role of normalization and particle size on particle-to-particle mass spectrum signal variability, comparing particles generated from solutions of 10 and 25 wt. % PEG-200. Top: particle-to-particle raw signal variability. Bottom: particle-to-particle normalized signal variability. Box-and-whisker plots are shown for each tracked m/z (corresponding to PEG-3 through PEG-7), with outliers defined as observations more than 1.5 times the interquartile range beyond the low and high quartiles. Individual observations are overlaid as points (distributed horizontally for clarity)."

**One gets the sense that the absolute signal variability is much greater than the 20-30% normalized signal variation reported by the authors.**

Yes, reducing the particle-to-particle variability in absolute signal is the reason we used a normalized signal, and we state this in the manuscript (§ 2.5, p. 6, line 22: "To account for particle-to-particle variability in MS signal, peaks were normalized to the PEG-6 parent ion signal at 283 m/z."). The figure we are adding to Appendix A will help give examples of the absolute and normalized signal variability (see above). We are continuing work to see if we

can reduce the amount of absolute signal variability, but we anticipate future experiments with this setup will continue to require an internal standard for quantification.

**Furthermore, it seems that the PEG droplet size studied in this experiment represents a practical lower limit, since working with a smaller size would probably have led to an unacceptably high fraction of particle spectra being filtered out from further analysis.**

It is true that with the setup described in the manuscript, it would become difficult to obtain sufficient signal in the mass spectra of smaller particles to quantify evaporation as we do here. However, we do not see this as a permanent limitation. We are working on improving the EDB-to-ionization transfer and the ionization source design so that smaller particles can be reliably quantified. We will expand on this point in the appropriate point in the Conclusion (§ 4, p. 12, line 15):

"Implementing an alternate ionization scheme could remove the limitation of only detecting molecules that are sufficiently volatile to vaporize quickly upon impact on the 220 °C platform. *Modification of the ionization scheme may also be necessary to obtain sufficient signal when working with smaller particles or analyte compounds present in smaller quantities.*"

**Section 2.2. Spring point measurements were made with 18 um diameter PMMA spheres. Is it possible that the existence of doublets and/or larger aggregates of PMMA spheres could have influenced the results?**

We will add the following to § 2.2 to explain why agglomerates did not influence the springpoint calibration (p. 4, line 25):

"The spring point of each sphere was measured for a number of different AC amplitude– frequency combinations, with a total of 22 spring point measurements over 4 different PMMA spheres. We ruled out the possibility of doublets or larger aggregates by observation of the droplet behavior by eye. Agglomerates show distinct scattering intensity fluctuations because of Brownian rotational motion in the EDB, which are easily detected by observing the image of the particle. Additionally, if the spring point had been measured using an aggregate with mass twice that of a single sphere or greater, the value would have been clearly anomalous and discarded."

**Section 3.2.1. "Particle" in line 13 of this page (fourth line of the section) should be plural.**

Corrected.

**References**

Bastelberger, S., Krieger, U. K., Luo, B. and Peter, T.: Diffusivity measurements of volatile organics in levitated viscous aerosol particles, Atmos. Chem. Phys., 17, 8453–8471, doi:10.5194/acp-17-8453-2017, 2017.

Cai, C., Stewart, D. J., Reid, J. P., Zhang, Y. H., Ohm, P., Dutcher, C. S. and Clegg, S. L.: Organic component vapor pressures and hygroscopicities of aqueous aerosol measured by optical tweezers, J. Phys. Chem. A, 119, 704–718, doi:10.1021/jp510525r, 2015.

Cotterell, M. I., Mason, B. J., Carruthers, A. E., Walker, J. S., Orr-Ewing, A. J. and Reid, J. P.: Measurements of the evaporation and hygroscopic response of single fine-mode aerosol particles using a bessel beam optical trap, Phys. Chem. Chem. Phys., 16, 2118–2128, doi:10.1039/c3cp54368d, 2014.

Dennis-Smither, B. J., Marshall, F. H., Miles, R. E. H., Preston, T. C. and Reid, J. P.: Volatility and oxidative aging of aqueous maleic acid aerosol droplets and the dependence on relative humidity, J. Phys. Chem. A, 118, 5680–5691, doi:10.1021/jp504823j, 2014.

Huisman, A. J., Krieger, U. K., Zuend, A., Marcolli, C. and Peter, T.: Vapor pressures of substituted polycarboxylic acids are much lower than previously reported, Atmospheric Chemistry and Physics, 13, 6647–6662, doi:10.5194/acp-13-6647-2013, 2013.

Krieger, U. K., Marcolli, C. and Reid, J. P.: Exploring the complexity of aerosol particle properties and processes using single particle techniques, Chem. Soc. Rev., 41, 6631–6662, doi:10.1039/c2cs35082c, 2012.

Krieger, U. K., Siegrist, F., Marcolli, C., Emanuelsson, E. U., Gøbel, F. M., Bilde, M., Marsh, A., Reid, J. P., Huisman, A. J., Riipinen, I., Hyttinen, N., Myllys, N., Kurtén, T., Bannan, T. and Topping, D.: A reference data set for validating vapor pressure measurement techniques: Homologous series of polyethylene glycols, Atmos. Meas. Tech. Discuss., 1–20, doi:10.5194/amt-2017-224, 2017.

Lienhard, D. M., Huisman, A. J., Bones, D. L., Te, Y.-F., Luo, B. P., Krieger, U. K. and Reid, J. P.: Retrieving the translational diffusion coefficient of water from experiments on single levitated aerosol droplets, Phys. Chem. Chem. Phys., 16, 16677–16683, doi:10.1039/C4CP01939C, 2014.

Mason, B. J., Cotterell, M. I., Preston, T. C., Orr-Ewing, A. J. and Reid, J. P.: Direct measurements of the optical cross sections and refractive indices of individual volatile and hygroscopic aerosol particles, J. Phys. Chem. A, 119, 5701–5713, doi:10.1021/acs.jpca.5b00435, 2015.

Rovelli, G., Miles, R. E. H., Reid, J. P. and Clegg, S. L.: Accurate measurements of aerosol hygroscopic growth over a wide range in relative humidity, J. Phys. Chem. A, 120, 4376–4388, doi:10.1021/acs.jpca.6b04194, 2016.

Stewart, D. J., Cai, C., Nayler, J., Preston, T. C., Reid, J. P., Krieger, U. K., Marcolli, C. and Zhang, Y. H.: Liquid–liquid phase separation in mixed organic/inorganic single aqueous aerosol droplets, J. Phys. Chem. A, 119, 4177–4190, doi:10.1021/acs.jpca.5b01658, 2015.

---

## Author Comment (AC3)

**Author Response to Anonymous Referee #3 of "Electrodynamic balance–mass spectrometry of single particles as a new platform for atmospheric chemistry research" by A.W. Birdsall et al.**

We thank the referee for their thoughtful comments, which have helped improve the manuscript. Our replies are below (referee comment in **bold**, response in normal face, new manuscript content in *italics*).

**A combination of electrodynamic trap and mass spectrometer has been described previously by (Dale et al., 1994), who used laser ablation to ionize the trapped particle. Please give a mention of this work in your review of analytical techniques.**

Thank you for the reference. We have added a reference to another paper from the same laboratory and era, which does use laser ablation to ionize a particle trapped directly in a modified ion trap of a mass spectrometer: Yang et al., *Anal. Chem.* 1995, 67, 1021-1025. (In Dale et al., 1994, the particles were dropped into the ion trap and ablated without being trapped.) We will add to the Introduction (p. 2, line 8):

"Among other features, the BQT design lends itself to study of condensed-phase reactions, triggered by the coalescence of two droplets of differing composition with sub-millisecond mixing times. *Additionally, in a different laboratory a quadrupole ion trap mass spectrometer was modified to levitate individual micron-sized droplets, followed by reducing the trap pressure over 20 minutes to ~0.1 Pa, ablating the particle with a pulsed laser (532 nm), and collecting a mass spectrum using the same ion trap (Yang et al., 1995).*"

**The possibility of studying the slow gas-phase chemical reactions in atmospheric aerosols is given as one of the motivations for this work. Aerosol particles, however, are much smaller in size (typically submicron) than the particles used in your work, so that trapping a particle in the EDB and transferring it into the ionization section might be very difficult. Could you discuss this issue in the outlook section? As a layman's suggestion, wouldn't it be possible to use a linear quadrupole trap to move a train of particles slowly through the reaction zone delivering them one by one into the ablation section coupled to the inlet of a mass spec? Such technique, though applied for optical particle characterization and not for mass spectrometry, has been recently described by (Sivaprakasam et al., 2017).**

We agree that trapping a submicron atmospheric aerosol particle would require further development. However, we would like to clarify that the immediate primary motivation for this setup is to study atmospherically-relevant processes in laboratory-generated aerosol in the micron size range. The goal is not to mimic atmospheric aerosol in all details, but to perform lab experiments which allow isolation and measurement of certain properties or processes. We can then apply what we learn from those measurements to submicron particles, accounting for size-dependent factors such as surface-to-volume ratio or possible diffusion limitations through the particle.

Returning to the question of trapping submicron aerosol particles, the linear quadrupole geometry you propose is certainly one possible approach. Recent work by Thomas Leisner and colleagues has also used a linear quadrupole to trap ensembles of particles, in their case nanoparticles with radius as small as 2 nm (Duft et al., 2015). With much less material in each particle, the detection limitations for the mass spectrometer would also need to be assessed.

Reflecting this discussion, we will add the following paragraph to § 4.1 ("Future experiments", p. 12, line 27):

"These strengths make the EDB-MS a complimentary technique to existing experimental and modeling approaches.

*Even using laboratory-generated aerosol particles with diameters on the order of 10 μm, results from future lab studies can be used to improve our understanding of submicron atmospheric aerosol particles. Physical and chemical constants, such as reaction rate constants and diffusion coefficients, are equally applicable to both laboratory and smaller atmospheric particles. The effect of other size-dependent factors, such as changing surface-to-volume ratio, radius-dependent mixing timescale of a viscous particle, or the Kelvin effect on growth of small nanoparticles, can be accounted for by calculation if the appropriate parameters are known. Trapping a submicron particle within the EDB-MS would require further development. One approach may be to transfer particles to the ionization region using a linear quadrupole geometry; this geometry has been used by other research groups (Duft et al., 2015; Jacobs et al., 2017; Sivaprakasam et al., 2017). Detection limitations for the mass spectrometer would also need to be assessed.*

One class of future experiments [...]"

**References**

Duft, D., Nachbar, M., Eritt, M. and Leisner, T.: A linear trap for studying the interaction of nanoparticles with supersaturated vapors, Aerosol Sci. Tech., 49, 683–691, doi:10.1080/02786826.2015.1063583, 2015.

Jacobs, M. I., Davies, J. F., Lee, L., Davis, R. D., Houle, F. and Wilson, K. R.: Exploring chemistry in microcompartments using guided droplet collisions in a branched quadrupole trap coupled to a single droplet, paper spray mass spectrometer, Anal. Chem., accepted, doi:10.1021/acs.analchem.7b03704, 2017.

Sivaprakasam, V., Hart, M. B. and Eversole, J. D.: Surface enhanced Raman spectroscopy of individual suspended aerosol particles, The Journal of Physical Chemistry C, 121, 22326–22334, doi:10.1021/acs.jpcc.7b05310, 2017.

Yang, M., Dale, J. M., Whitten, W. B. and Ramsey, J. M.: Laser desorption mass spectrometry of a levitated single microparticle in a quadrupole ion trap, Anal. Chem., 67, 1021–1025, doi:10.1021/ac00102a001, 1995.

---

## Author Response (AR1)

**Author Response for "Electrodynamic balance–mass spectrometry of single particles as a new platform for atmospheric chemistry research" by A.W. Birdsall et al.**

We thank the referees for their thoughtful comments, which have helped improve the manuscript. Our replies are below (referee comment in **bold**, response in normal face, new manuscript content in *italics*, removed manuscript content in ).

**Anonymous Referee #1**

**The authors might consider including a sentence or two on the use of optical traps since these are also used with great success to study atmospherically relevant particles.**

We agree that optical traps are a highly valuable platform for single particle studies. The introduction's treatment of levitated droplet experiments does acknowledge that optical, electrodynamic, and acoustic traps have all been used to study levitated particles. To clarify and emphasize the important contributions of both EDBs and optical traps, we will amend the Introduction as follows (p. 1, line 27):

"Researchers have studied particles deposited onto a substrate, or alternately, particles levitated by means of a "trapping" force originating from an electric field, radiation pressure of a laser beam, or acoustic waves. Levitated droplet experiments are appealing because they mimic aerosol particles in the ambient environment in certain key ways: the presence of a surrounding bath gas, an enhanced surface-to-bulk ratio, the absence of physical contact with a substrate, and the ability to study supersaturated particles.  *Using electrodynamic or optical forces, multiple laboratories have analyzed levitated droplets using optical techniques such as Raman spectroscopy and Mie resonance spectroscopy (earlier work reviewed in Krieger et al., 2012). A number of different properties have been studied in this way, including vapor pressures (Cai et al., 2015; Cotterell et al., 2014; Huisman et al., 2013; Krieger et al., 2017), hygroscopic growth (Cai et al., 2015; Cotterell et al., 2014; Rovelli et al., 2016), optical properties (Mason et al., 2015), liquid-liquid phase separation (Stewart et al., 2015), diffusivities and diffusion coefficients (Bastelberger et al., 2017; Lienhard et al., 2014), and oxidative aging (Dennis-Smither et al., 2014)."*

**The topic of droplet size and the ion signal intensity produced from it should be discussed more coherently in the manuscript.**

We thank the referee for a number of suggestions to clarify the manuscript on the topic of droplet size and ion signal intensity. We address the individual points below.

**The authors mention 10-30 micrometers in the introduction, but it is not clear whether this is limited by the balance or mass spectrometer.**

The EDB can operate with smaller diameters; the operating diameter was chosen for a number of other factors (particle injection, ease of transfer to ionization region, magnitude of MS signal). We will clarify discussion of size range in the Introduction with the following change (p. 2, line 28):

"Here we describe a newly-developed system that couples an electrodynamic balance (EDB), which levitates aerosol particles  for an arbitrarily long amount of time, with mass spectral analysis of the entire particle. *We operate with particles of diameter approximately 10–30 µm for reliable acquisition of quantifiable mass spectra, though the EDB can levitate particles of smaller diameter.*"

**[...] droplet size and/or analyte mass sampled is not given in the discussion or captions of Figures 2 and 3.**

We will clarify in the Figure 2 and 3 captions and surrounding body text that the figures are from a 20 µm diameter particle:

§ 2.5 (p. 6, line 20): "Fig. 2 presents a sample time trace of selected ion signals arising from ejection and ionization of a *20 µm diameter* PEG-200 particle."

§ 3.1 (p. 8, line 4): "A sample mass spectrum of a PEG-200 particle (Fig. 3) shows that the signal from  *droplets with diameters on the order of 20 µm* can be easily detected."

Fig. 2 caption (p. 16, line 3): "Here the time series corresponding to a mass spectrum of Fig. 3, of a single PEG-200 droplet *with diameter 20 µm*, is shown.

Fig. 3 caption (p. 17, line 2): "Sample mass spectrum of a droplet consisting of polyethylene glycol, average molar mass 200 g mol$^{-1}$ (PEG-200), *droplet diameter 20 µm*.

**Nor is it discussed in any detail Section 3.4, which otherwise gives an excellent discussion of particle-to-particle signal variability.**

For particles used in the evaporation experiments the variability in measured diameters, using the spring-point technique, was found not to explain the variability in normalized signal as evaporation took place. In other words, the particle-to-particle variability was dominated by factors other than particles with ~9 µm radius evaporating faster than particles with a ~10 µm radius. This is what was meant in the manuscript by the statement, "The variability was not explained by the variability in measured starting particle diameter." (§ 3.4, p. 10, line 13)

We have since performed additional checks on the role of size variation in ion signal intensity and variability:

- We considered PEG-200 particles which were trapped and "immediately" ejected from the EDB (within 6 minutes of introduction), that were measured with the spring-point technique, and met the threshold of having sufficient signal (defined as at least 1000

counts at the m/z 283, PEG-6 channel). For these particles, there was no correlation between the particle diameter and raw signal. There is insufficient data to determine whether within this size range there is any systematic difference in the amount of particle-to-particle signal variability.

- We have other data consisting of mass spectra for PEG-200 particles generated from solutions of varying weight fractions of PEG-200 in water. The weight fraction of PEG-200 in water can be taken as proportional to the starting trapped particle mass after fast water evaporation, since particle diameters were not measured with the spring-point technique. We compared particles generated from 10 wt% and 25 wt% PEG-200 mixtures, again restricting to particles immediately ejected from the EDB and over the signal threshold (1000 counts at m/z 283). The particle-to-particle raw signal variability was less for droplets from the 25 wt% mixture, but the normalized signal variability was similar for droplets from the two mixtures.

To clarify and summarize this analysis, we will update § 3.4 and Appendix A, including a new figure and caption:

§ 3.4 (p. 10, line 13): "We investigated a number of possible factors contributing to the variability in signal. *Within the size range of particles analyzed during the evaporation experiments, variability in the particle diameter (approx. ±10%), measured with the spring-point method, did not correlate to particle-to-particle variability in apparent evaporation rates, or to particle-to-particle variability in absolute signal. For two populations of PEG-200 particles with masses varying by a factor of approximately 2.5, higher variability in raw signal was observed for the smaller particles. In this data set the decreased particle-to-particle variability in normalized signal can also be readily observed (Fig. A1). From these analyses we conclude that though the particle-to-particle variability in raw signal may be affected by significant differences in particle mass, the*  variability *in the normalized evaporation data* was not explained by the variability in measured starting particle diameter."

Appendix A:

p. 13, line 29: "Appendix A: Characterization of mass spectrometer fragmentation,  sensitivity*, and signal variability*"

p. 13, line 30: "The appendix includes data on PEG mass spectral relative sensitivities (Table A1) and fragmentation patterns (Table A2). *The role of normalization and particle size on particle-to-particle mass spectrum signal variability was analyzed (Fig. A1). Mass spectra for a set of PEG-200 particles were considered: those for particles "immediately" ejected from the EDB after trapping (defined as within 6 minutes of introduction), and with spectra over the signal threshold (defined as at least 1000 counts in the m/z 283, PEG-6 channel). The particles were generated from two different solutions of PEG-200 in water: 10% and 25% by weight. After fast water evaporation, the weight fraction can be taken as proportional to the starting trapped particle mass. (A full set of spring-point diameter measurements are unavailable.) The particle-to-particle raw signal variability (Fig. A1, top) was less for droplets from the 25 wt. % mixture, but the normalized signal variability (Fig. A1,*

*bottom) was similar for droplets from the two mixtures. In both cases, the variability in the normalized signal is smaller than the variability in the raw signal."*

[...]

[Figure]

*"Figure A1: Analysis of the role of normalization and particle size on particle-to-particle mass spectrum signal variability, comparing particles generated from solutions of 10 and 25 wt. % PEG-200. Top: particle-to-particle raw signal variability. Bottom: particle-to-particle normalized signal variability. Box-and-whisker plots are shown for each tracked m/z (corresponding to PEG-3 through PEG-7), with outliers defined as observations more than 1.5 times the interquartile range beyond the low and high quartiles. Individual observations are overlaid as points (distributed horizontally for clarity)."*

**One gets the sense that the absolute signal variability is much greater than the 20-30% normalized signal variation reported by the authors.**

Yes, reducing the particle-to-particle variability in absolute signal is the reason we used a normalized signal, and we state this in the manuscript (§ 2.5, p. 6, line 22: "To account for particle-to-particle variability in MS signal, peaks were normalized to the PEG-6 parent ion

signal at 283 m/z."). The figure we are adding to Appendix A will help give examples of the absolute and normalized signal variability (see above). We are continuing work to see if we can reduce the amount of absolute signal variability, but we anticipate future experiments with this setup will continue to require an internal standard for quantification.

**Furthermore, it seems that the PEG droplet size studied in this experiment represents a practical lower limit, since working with a smaller size would probably have led to an unacceptably high fraction of particle spectra being filtered out from further analysis.**

It is true that with the setup described in the manuscript, it would become difficult to obtain sufficient signal in the mass spectra of smaller particles to quantify evaporation as we do here. However, we do not see this as a permanent limitation. We are working on improving the EDB-to-ionization transfer and the ionization source design so that smaller particles can be reliably quantified. We will expand on this point in the appropriate point in the Conclusion (§ 4, p. 12, line 15):

"Implementing an alternate ionization scheme could remove the limitation of only detecting molecules that are sufficiently volatile to vaporize quickly upon impact on the 220 °C platform. *Modification of the ionization scheme may also be necessary to obtain sufficient signal when working with smaller particles or analyte compounds present in smaller quantities.*"

**Section 2.2. Spring point measurements were made with 18 um diameter PMMA spheres. Is it possible that the existence of doublets and/or larger aggregates of PMMA spheres could have influenced the results?**

We will add the following to § 2.2 to explain why agglomerates did not influence the spring-point calibration (p. 4, line 25):

"The spring point of each sphere was measured for a number of different AC amplitude–frequency combinations, with a total of 22 spring point measurements over 4 different PMMA spheres. *We ruled out the possibility of doublets or larger aggregates by observation of the droplet behavior by eye. Agglomerates show distinct scattering intensity fluctuations because of Brownian rotational motion in the EDB, which are easily detected by observing the image of the particle. Additionally, if the spring point had been measured using an aggregate with mass twice that of a single sphere or greater, the value would have been clearly anomalous and discarded.*"

**Section 3.2.1. "Particle" in line 13 of this page (fourth line of the section) should be plural.**

Corrected.

**Anonymous Referee #2**

**Other groups have reported using a paper-spray electrospray ionization (Jacobs et al.) and HV needle desorption (Tracey et al.) , I can also remember seeing some**

results using DART and single droplet analysis. I hope we will soon see some comparisons between these ionization techniques. Molecular dependent ionization and detection efficiency will always be a key issue.

**This paper does well-discuss and highlight the issues around quantitative MS sampling of droplets - particularly when "single droplet" mass spectra are distinguished.**

**For single microdroplets, droplet-to-droplet variations affect desorption and ionization efficiency and - more insidiously - can affect ions differently. This is the well-known challenge with just about all forms of quantitative mass spectrometry. This paper discusses these concerns towards the end of the results section very well.**

We thank the referee for their comments. We agree that both the Jacobs et al. (2017) and Tracey et al. (2014) approaches, which we cite in the manuscript, are also valuable, and agree that a careful experimental comparison would be valuable, though it is outside of the scope of the present manuscript. We further agree that quantitative interpretation of our results required both careful calibration of the signal response, as well as averaging over droplet-to-droplet variations. Work is underway in our laboratory to optimize the setup to reduce droplet-to-droplet variations, so that less averaging over multiple droplets is required.

**Anonymous Referee #3**

**A combination of electrodynamic trap and mass spectrometer has been described previously by (Dale et al., 1994), who used laser ablation to ionize the trapped particle. Please give a mention of this work in your review of analytical techniques.**

Thank you for the reference. We have added a reference to another paper from the same laboratory and era, which does use laser ablation to ionize a particle trapped directly in a modified ion trap of a mass spectrometer: Yang et al., *Anal. Chem.* 1995, 67, 1021-1025. (In Dale et al., 1994, the particles were dropped into the ion trap and ablated without being trapped.) We will add to the Introduction (p. 2, line 8):

"Among other features, the BQT design lends itself to study of condensed-phase reactions, triggered by the coalescence of two droplets of differing composition with sub-millisecond mixing times. *Additionally, in a different laboratory a quadrupole ion trap mass spectrometer was modified to levitate individual micron-sized droplets, followed by reducing the trap pressure over 20 minutes to ~0.1 Pa, ablating the particle with a pulsed laser (532 nm), and collecting a mass spectrum using the same ion trap (Yang et al., 1995).*"

**The possibility of studying the slow gas-phase chemical reactions in atmospheric aerosols is given as one of the motivations for this work. Aerosol particles, however, are much smaller in size (typically submicron) than the particles used in your work, so that trapping a particle in the EDB and transferring it into the ionization section might be very difficult. Could you discuss this issue in the outlook section? As a layman's suggestion, wouldn't it be possible to use a linear quadrupole trap to move**

**a train of particles slowly through the reaction zone delivering them one by one into the ablation section coupled to the inlet of a mass spec? Such technique, though applied for optical particle characterization and not for mass spectrometry, has been recently described by (Sivaprakasam et al., 2017).**

We agree that trapping a submicron atmospheric aerosol particle would require further development. However, we would like to clarify that the immediate primary motivation for this setup is to study atmospherically-relevant processes in laboratory-generated aerosol in the micron size range. The goal is not to mimic atmospheric aerosol in all details, but to perform lab experiments which allow isolation and measurement of certain properties or processes. We can then apply what we learn from those measurements to submicron particles, accounting for size-dependent factors such as surface-to-volume ratio or possible diffusion limitations through the particle.

Returning to the question of trapping submicron aerosol particles, the linear quadrupole geometry you propose is certainly one possible approach. Recent work by Thomas Leisner and colleagues has also used a linear quadrupole to trap ensembles of particles, in their case nanoparticles with radius as small as 2 nm (Duft et al., 2015). With much less material in each particle, the detection limitations for the mass spectrometer would also need to be assessed.

Reflecting this discussion, we will add the following paragraph to § 4.1 ("Future experiments", p. 12, line 27):

"These strengths make the EDB-MS a complimentary technique to existing experimental and modeling approaches.

*Even using laboratory-generated aerosol particles with diameters on the order of 10 μm, results from future lab studies can be used to improve our understanding of submicron atmospheric aerosol particles. Physical and chemical constants, such as reaction rate constants and diffusion coefficients, are equally applicable to both laboratory and smaller atmospheric particles. The effect of other size-dependent factors, such as changing surface-to-volume ratio, radius-dependent mixing timescale of a viscous particle, or the Kelvin effect on growth of small nanoparticles, can be accounted for by calculation if the appropriate parameters are known. Trapping a submicron particle within the EDB-MS would require further development. One approach may be to transfer particles to the ionization region using a linear quadrupole geometry; this geometry has been used by other research groups (Duft et al., 2015; Jacobs et al., 2017; Sivaprakasam et al., 2017). Detection limitations for the mass spectrometer would also need to be assessed.*

One class of future experiments [...]"

**Miscellaneous updates**

We will move a majority of the caption text for Fig. B1 out of the caption and into the body of Appendix B, to match the style of Fig. A1:

Appendix B (p. 14, line 2):

[revised manuscript text omitted]